# Response of Soil Microenvironment and Crop Growth to Cyclic Irrigation Using Reclaimed Water and Brackish Water

**DOI:** 10.3390/plants12122285

**Published:** 2023-06-12

**Authors:** Chuncheng Liu, Juan Wang, Pengfei Huang, Chao Hu, Feng Gao, Yuan Liu, Zhongyang Li, Bingjian Cui

**Affiliations:** 1Institute of Farmland Irrigation, Chinese Academy of Agricultural Sciences, Xinxiang 453002, China; liuchuncheng986@sohu.com (C.L.); huangpengfei@caas.cn (P.H.); huchao@caas.cn (C.H.); gaofengcaas@sina.com (F.G.); liuyuanfiri88@163.com (Y.L.); 2Key Laboratory of High-Efficient and Safe Utilization of Agriculture Water Resources, Chinese Academy of Agricultural Sciences, Xinxiang 453002, China; 3Agriculture Water and Soil Environmental Field Science Research Station of Xinxiang City, Chinese Academy of Agricultural Sciences, Xinxiang 453000, China; 4College of Hydraulic Science and Engineering, Yangzhou University, Yangzhou 225000, China; blesswangj@163.com

**Keywords:** antioxidation property, brackish water, reclaimed water, cyclic irrigation, secondary soil salinization

## Abstract

The scarcity of freshwater resources has increased the use of nonconventional water resources such as brackish water, reclaimed water, etc., especially in water-scarce areas. Whether an irrigation cycle using reclaimed water and brackish water (RBCI) poses a risk of secondary soil salinization to crop yields needs to be studied. Aiming to find an appropriate use for different nonconventional water resources, pot experiments were conducted to study the effects of RBCI on soil microenvironments, growth, physiological characteristics and antioxidation properties of crops. The results showed the following: (1) compared to FBCI, the soil moisture content was slightly higher, without a significant difference, while the soil EC, sodium and chloride ions contents increased significantly under the RBCI treatment. With an increase in the reclaimed water irrigation frequency (Tri), the contents of EC, Na^+^ and Cl^−^ in the soil decreased gradually, and the difference was significant; the soil moisture content also decreased gradually. (2) There were different effects of the RBCI regime on the soil’s enzyme activities. With an increase in the Tri, the soil urease activity indicated a significant upward trend as a whole. (3) RBCI can alleviate the risk of soil salinization to some extent. The soil pH values were all below 8.5, and were without a risk of secondary soil alkalization. The ESP did not exceed 15 percent, and there was no possible risk of soil alkalization except that the ESP in soil irrigated by brackish water irrigation went beyond the limit of 15 percent. (4) Compared with FBCI, no obvious changes appeared to the aboveground and underground biomasses under the RBCI treatment. The RBCI treatment was conducive to increasing the aboveground biomass compared with pure brackish water irrigation. Therefore, short-term RBCI helps to reduce the risk of soil salinization without significantly affecting crop yield, and the irrigation cycle using reclaimed-reclaimed-brackish water at 3 g·L^−1^ was recommended, according to the experimental results.

## 1. Introduction

Drought has a negative impact on the growth of crops and the quality of fruits, especially in water-scarce areas [1,2]. The scarcity of freshwater (FW) resources has increased the use of brackish water (BW), especially in water-scarce areas [3], in order to sustain agriculture for sustainable development. However, long-term BW irrigation may increase soil salinity, bulk density and water content [4,5,6], resulting in soil salinization, which has adverse effects on soil physicochemical characteristics and nutrient transformation. As much as 20% of the world’s arable land and 33% of irrigated farmland are impacted by high salinity [7]. In addition, due to several primary causes, including low rainfall, strong surface evaporation, primary rock weathering, saline water irrigation and poor farming practices, salinized areas are growing at an annual rate of 10%; It is estimated that more than 50% of arable land will be salinized by 2050 [7]. Soil salinity is one of the most destructive environmental stresses, resulting in a significant decline in cultivated land area, crop productivity and quality [8,9], which is the main limiting factor of the agricultural economy in arid areas [10]. As an alternative source of FW, unconventional water will be widely used in agriculture in the near future. Water scarcity in arid areas is driving agriculture to make greater use of marginal or inferior water sources for irrigation [11], as unconventional water (such as BW (2–5 g·L^−1^), saline water (5–8 g·L^−1^) or reclaimed water (RW)) is increasingly used to irrigate crops [12,13]. Therefore, it is vital for agricultural production to study the safe utilization of unconventional water resources.

Studies have shown that RW- or BW-irrigated cotton is viable [14,15]. To some extent, BW irrigation can promote the growth of crops, maintain yield, and also improve water use efficiency [16]. Although BW is rich in beneficial micronutrients, BW irrigation may also result in toxic stress to plant growth through enriching some ions, such as Na^+^, Cl^−^ and HCO_3_^−^ [17,18]. Saline water contains a lot of sodium ions, which improves the exchangeable sodium content in soil and causes soil particles to fragment, thus increasing the content of dispersed clay particles [19]. While alleviating crop water stress, saline water irrigation also can cause soil salt accumulation, especially root salt accumulation, which may have an adverse impact on crop yield through osmosis and ion poisoning [20]. Results have shown that short-term BW irrigation has no significant impacts on soil chemical properties and soil salinization, but long-term BW irrigation may lead to secondary soil salinization [21], soil water repellency [22] and crop growth [23]. The co-use of BW and FW can alleviate secondary soil salinization and the impacts of salt in BW on the growth of crops [24]. For example, the desalination effect in crops’ root-dense area (5~45 cm) under intermittent combined irrigation is better, and can provide a better growing environment for crops [25]. In view of the characteristics of abundant BW resources and its low utilization rate in northern Shandong Province, an irrigation cycle system was established under returning rice straw to the field, and the annual water–salt balance safe efficient technical system was formed, including freshwater leaching salt, supplementary irrigation with saline water, returning rice straw to the field to inhibit salt and precipitation leaching salt [26]. In addition, cyclic irrigation using BW and FW was conducive to the accumulation of soil carbon and the maintenance of the soil’s nutrient pool [27]. However, in areas where freshwater resources are scarce, the utilization of BW is limited to a great extent, and new utilization approaches need to be explored. As a kind of unconventional water resource, reclaimed water is rich, and has a lower salt content than BW. The research objects of reclaimed water utilization mainly focus on crop growth [28], quality [29], soil environment [30], soil microbial community structure [31], underground water [32], suitable irrigation technology [33] and so on. The results showed that under 15-year RW irrigation, the concentration of trace elements on leaf surfaces did not exceed the threshold, and the concentrations of trace elements in parks irrigated by RW for 10 years did not obviously differentiate from those parks irrigated without reclaimed water [34]. RW irrigation did not affect the concentrations of heavy metals and trace elements in leaves and fruits (e.g., Na, B, Zn), and it was viable to use RW to irrigate [35]. Romero-Trigueros et al. also demonstrated the medium-and long-term feasibility of RW irrigation for citrus [36]. In theory, the salinity in RW is less than that in BW, so irrigation with RW may play a role in leaching salt and avoiding secondary soil salinization to a certain extent. We hypothesized that RW could replace FW to irrigate crops with BW alternatively. Our objective was to identify the irrigation cycle mode of using BW and RW based on crop characteristics and soil environmental indictors, and to provide some theoretical guidance for marginal quality water use for agricultural irrigation in freshwater-scarce areas.

## 2. Results

### 2.1. Variations in Physical and Chemical Properties of Soil under Cyclic Irrigation

#### 2.1.1. Soil Moisture Content and EC

As seen in Figure 1, the soil moisture content in R improved slightly by 1.10% compared to F, and the difference was not significant. The soil moisture content under an irrigation cycle using reclaimed water and brackish water (RBCI) was slightly higher without significant difference compared with FBCI. Under RBCI, as the irrigation times of RW increased, the soil moisture content decreased gradually on the whole; the soil moisture contents in the cyclic irrigation treatments were 16~24% lower than that in the BW irrigation treatment, and the differences reached a significant level. For the same irrigation sequence, the soil moisture content positively correlated with the salinity, and there was a significant difference between B3 and B5. Therefore, there was no significant difference in the soil moisture content between R and F, as well as between RBCI and FBCI. Meanwhile, the soil moisture content tended to decrease as the irrigation times of RW increased.

The soil EC in R was 817.67 μS·cm^−1^ and significantly increased by 49.6% compared with F. The soil EC under RBCI increased significantly overall compared to soils under FBCI, with an increase of 4.97~18.35%. Under RBCI, at the same salinity in BW, the soil’s EC declined gradually as the irrigation times with RW increased, and the difference reached an obvious level except that there was no significant difference between RB3 and RRB3. In addition, compared to BW irrigation (B3 and B5), the soil’s EC under cyclic irrigation treatments decreased by 22~39%. With the same irrigation sequence, the soil’s EC was positively correlated with the salinity, and no significant difference existed between RRB3 and RRB5.

#### 2.1.2. Contents of Water-Soluble Na^+^ and Cl^−^ in Soil

As seen from Table 1, the soil’s Na^+^ and Cl^−^ contents in R were obviously higher than those in F, with increases of 438.17% and 50.96%, respectively. Compared with FBCI, the soil Na^+^ and Cl^−^ contents showed increasing trends under RBCI, and the difference in soil Na^+^ content reached a significant level. Under RBCI, at the same salinity in BW, the soil’s Na^+^ and Cl^−^ contents declined with the increase in irrigation times with RW, and the cyclic irrigation treatments were significantly lower than the BW irrigation treatment, with decreases of 48~61% and 36~54%, respectively. With the increase in salinity in BW, the contents of soil Na^+^ and Cl^−^ increased gradually, and the difference was significant (except for the difference between RRB3 and RRB5).

#### 2.1.3. SOM and WDPT of Soil

As seen in Figure 2, (1) no obvious difference was observed in the SOM between R and F. Under RBCI, the SOM content showed an increasing trend, and the SOM in RB3 was obviously higher than that in FB3 compared to FBCI. Under RBCI irrigation, as the times of RW increased, the SOM showed a downward trend, but the difference did not reach a significant level (except RRB5 was significantly lower than B5). For the same irrigation sequence, the SOM increased slightly as the salinity in BW increased, but the difference did not reach an obvious level.

(2) The WDPT in R was 5.21 s, representing a weak water repellency that was significantly higher than that in F. Compared with FBCI, the WDPT showed no significant difference under RBCI. Under RBCI, the WDPT in cyclic irrigation was the lowest; as the salinity in BW increased, the WDPT showed no obvious changes.

### 2.2. Soil Enzyme Activity

As seen from Table 2, the activities of S-AKP/ALP, S-SC and S-UE in R were 16.49%, 1.30% and 8.88% higher than those in F, respectively. Compared to FBCI, there was no significant difference in S-AKP/ALP activity except that RB3 was significantly higher than FB3, with no significant difference in S-SC activity except that RRB3 was significantly higher than FFB3 under RBCI. The S-UE activity in “reclaimed-brackish water” cyclic irrigation was significantly lower than that under the “freshwater-brackish water” cyclic irrigation treatment, while S-UE activity under “reclaimed-reclaimed-brackish water” cyclic irrigation was higher than that of “freshwater-freshwater-brackish water” cyclic irrigation as a whole.

Under RBCI treatment, the S-UE activity increased significantly, and the difference reached a significant level as the RW irrigation times increased, while the S-SC and S-AKP/ALP activity decreased generally, and had no significant differences (except that the S-SC activity in RB3 was significantly lower than that in RRB3). For the same irrigation sequence, the activities of S-AKP/ALP, S-SC and S-UE increased with the increase in salinity with BW, except that the activity of S-SC in RRB5 was significantly lower than that in RRB3.

### 2.3. Risk Analysis of Secondary Soil Salinization

As seen in Figure 3, (1) the soil pH was 7.91 in R, which was 1.54% higher than that in F, and the difference was significant. Compared with FBCI, the soil pH increased under RBCI, and the difference reached a significant level at a salinity of 5 g·L^−1^ in BW.

Under the RBCI treatment, the soil pH in the cyclic irrigation treatment was generally higher than that of BW irrigation, and the difference was significant at low BW salinity (3 g·L^−1^). For the same irrigation sequence, with the increase of salinity in BW, the soil pH in the cyclic irrigation treatments increased, and the difference reached a significant level, except that the difference was not significant between RB3 and RB5.

(2) The soil-exchangeable K/Na in R was 0.51, and was 5.56% lower than that in F, but the difference did not reach a significant level. Compared with FBCI, the soil-exchangeable K/Na decreased under RBCI, and the difference reached a significant level.

Under the RBCI treatment, at the same salinity in BW, the soil-exchangeable K/Na increased as the RW irrigation times increased, and the difference reached a significant level. At the same irrigation sequence, the soil-exchangeable K/Na decreased as the salinity in BW increased, and the difference was significant.

(3) The soil ESP in R was 5.42%, which was much lower than the soil salinization threshold (15%), and there was no risk of soil salinization. The soil ESP in R was 46.31% higher than that in F, and the difference reached a significant level. Compared with FBCI, the soil ESP showed an increasing trend under RBCI, and the difference was significant.

Under the RBCI treatment, the soil ESP decreased as the RW irrigation times increased, and the difference reached a significant level. For the same irrigation sequence, the soil ESP increased with the increase in salinity in BW, and the difference was significant except the difference between RRB3 and RRB5.

In addition, the ESP in B3 and B5 were both more than 15% with a certain risk of alkalization, while the ESP in other treatments all did not exceed 15% without the risk of possible soil alkalization.

### 2.4. Physiological and Growth Characteristics of Crops

#### 2.4.1. Biomass of Crops

As shown in Figure 4, AFW and ADW in R were 7.07% and 5.25% higher than that in F, respectively, but no significant difference appeared. Compared with FBCI, AFW and ADW under RBCI increased slightly at low BW salinity (3 g·L^−1^), and decreased slightly at high BW salinity (5 g·L^−1^); however, the difference was not significant. Under the RBCI treatment, the crops irrigated by reclaimed water had the highest AFW and ADW, and the crops’ AFW and ADW under cyclic irrigation were higher than those under BW irrigation, with increases of 7~30% and 23~43%, respectively. Under the same irrigation sequence, AFW and ADW declined as the salinity in BW increased, but no significant difference existed.

Although UFW and UDW in R increased by 24.29% and 11.40% compared with F, respectively, no significant difference existed. Compared with FBCI, no significant difference existed in UFW and UDW under RBCI. Under RBCI, at the same salinity level in BW, the UFW and UDW in cyclic irrigation treatments were higher than those in the BW irrigation treatment, with increases of 12–42% and 0–22%, respectively; the highest UFW and UDW appeared in RB3/RB. For the same irrigation sequence, there was no significant difference in UFW and UDW with the increase in salinity in BW.

#### 2.4.2. Chlorophyll Content

As seen from Table 3, compared with F, the contents of chlorophyll a, chlorophyll b and total chlorophyll in R decreased by 12.43%, −7.26% and 8.54%, respectively, but no significant difference existed. Compared with FBCI, the chlorophyll a and total chlorophyll contents both showed an increasing trend, while the chlorophyll b content showed a significantly increasing trend at low salinity (3 g·L^−1^), and decreased at high BW salinity without a significant difference under RBCI.

Under the RBCI treatment, at the same salinity in BW, the highest chlorophyll a and total chlorophyll leaf contents appeared in RB3/RB5, but the difference between the treatments was not significant. The chlorophyll b content in RRB3/RRB5 was significantly higher than those of other treatments at low BW salinity (3 g·L^−1^), but the highest chlorophyll b content appeared in RB3/RB5 at high BW salinity (5 g·L^−1^) without a significant difference between treatments. For the same irrigation sequence, with the increase in salinity in BW, there was no significant difference in chlorophyll a and total chlorophyll leaf contents, but the chlorophyll b content decreased significantly in the cyclic irrigation treatments.

### 2.5. Antioxidant Characteristics of Crops

#### 2.5.1. Antioxidant Enzymes of Leaves

Compared to F, the CAT and POD activities in R decreased by 42.23% and 22.45%, respectively, without significant difference, while the SOD activity significantly increased by 3.39 times (Figure 5). Compared with FBCI, the CAT activity increased slightly under the RBCI treatment, but the difference was not significant. The SOD activity increased significantly on the whole, except that the difference was not obvious between RRB5 and FFB5; the POD activity decreased slightly without a significant difference.

Under the RBCI treatment, at the same salinity in BW, the leaf CAT activity decreased slightly, and the SOD activity increased significantly with an increase in reclaimed water irrigation times; however, the POD activity in the cyclic irrigation treatments was inhibited to a certain extent compared with the BW irrigation treatments. For the same irrigation sequence, the activities of CAT, SOD and POD decreased slightly with the increase in salinity in BW, but there was no significant difference between the treatments.

#### 2.5.2. MDA

The MDA content in R increased by 20.36% compared to F without a significant difference. Compared with FBCI, the MDA content had no obvious change except for RB3 under RBCI. Under RBCI, at the same salinity in BW, the MDA content in RRB3/RRB5 was the lowest. At the same irrigation sequence, there was no significant difference in the MDA content between treatments as the salinity in BW increased.

#### 2.5.3. Soluble Protein Content

As seen in Figure 6, the soluble protein content in R increased by 25.33% compared to F without a significant difference. Compared with FBCI, the soluble protein content under RBCI increased at low salinity (3 g·L^−1^) in BW, but decreased at high salinity (5 g·L^−1^) in BW, with a significant difference. Under the RBCI treatment, at the same salinity in BW, the soluble protein content had no obvious change as the RW irrigation times increased. At the same irrigation sequence, the higher the salinity in BW, the lower the soluble protein content, except for those under BW irrigation, and no significant difference existed between the treatments.

### 2.6. Distribution of Na^+^ in Soil–Crop

#### 2.6.1. Na^+^ Content in Soil and Leaves

As seen from Table 4, (1) the soil Na^+^ content in R was 4.38 times higher than that in F. The soil Na^+^ contents under FBCI were significantly higher than those under RBCI, except there was no significant difference between RB5 and FB5. Under RBCI, the soil Na^+^ content declined as the RW irrigation times increased; at the same irrigation sequence, the higher the salinity, the higher the soil Na^+^ content, except that no significant difference was found between RRB3 and RRB5.

(2) The Na^+^ content of leaves in R increased by 46.40% compared with F, and the difference reached a significant level. Compared with FBCI, the leaf Na^+^ content decreased under RBCI, and there was a significant difference between RB3 and FB3, as well as between RRB5 and FFB5. Under RBCI, the leaf Na^+^ content decreased with an increase in RW irrigation times, and the difference reached a significant level except for the difference between RB3 and RRB3. At the same irrigation sequence, the higher the salinity in BW, the higher the leaf Na^+^ content without a significant difference between treatments, except for the difference between B3 and B5. Therefore, compared with BW irrigation, cyclic irrigation can obviously reduce the absorption of Na^+^ content by the leaves.

#### 2.6.2. Accumulation of Na^+^ in Soil and Na^+^ Uptake Efficiency of Leaves

According to the soil Na^+^ content and the initial soil Na^+^ content, the Na^+^ accumulations in soil were calculated, and the results are shown in Figure 7. The change trend of Na^+^ accumulation in soil was consistent with that of the soil’s Na^+^ content, and there was no accumulation of Na^+^ in soil in F.

According to the Na^+^ content in leaves, crop biomass and soil Na^+^ accumulation, the total Na^+^ input could be calculated, and then the Na^+^ uptake efficiency of leaves could be calculated, as shown in Figure 8.

As seen in Figure 8, compared with FBCI, the Na^+^ absorption efficiency by the leaves tended to decrease under RBCI, with a significant difference at high BW salinity (5 g·L^−1^). Therefore, cyclic irrigation may reduce the efficiency of Na^+^ absorption by leaves. Under RBCI, the Na^+^ absorption efficiency by leaves generally increased on the whole as the RW irrigation times increased; at the same irrigation sequence, the higher the salinity, the higher the Na^+^ absorption efficiency by leaves without a significant difference. Therefore, an increase in irrigation times with reclaimed water may promote the Na^+^ absorption efficiency of leaves.

## 3. Discussion

### 3.1. Response of Soil Physicochemical Properties to RBCI

Some reports have proven that reclaimed water or saline water could be reasonably used for agricultural irrigation. However, long-term irrigation may lead to soil salt accumulation [37], resulting in an increase in soil bulk density [38]. In addition, excessive utilization of BW can lead to the accumulation of salt and toxic ions, which can have a negative impact on the mineralization process, and lead to the reduced availability of essential nutrients [39]. The results in this study showed that the soil water content and EC under BW irrigation were significantly higher than those under freshwater irrigation, and the soil water and salt contents under RBCI were also higher than those under FBCI (Figure 1). This is due to the salinity in BW reducing soil water potential and causing crops to suffer from saline–alkali stress, which has a negative impact on root water uptake, thus increasing the soil moisture content [16]. The accumulation of Na^+^ and Cl^−^ can cause osmotic and ion stress in plants, causing cytotoxic effects [40,41]. High concentrations of sodium ions have toxic effects on cell metabolism, inhibiting enzyme activity, cell division and expansion, leading to irregular cell membrane and osmotic imbalance, and restraining the growth of crops [42]. Our results showed that the contents of soil Na^+^ and Cl^−^ under RW irrigation and BW irrigation were both significantly higher than those under FW irrigation treatment, and the soil Na^+^ and Cl^−^ contents under RBCI showed an increasing trend compared with FBCI (Table 1); this mainly depended on the ion concentrations in the irrigation water. However, compared with BW irrigation, the contents of Na^+^ and Cl^−^ in soil under cyclic irrigation decreased significantly.

In this research, the SOM under BW irrigation was higher than that under freshwater irrigation (Figure 2), and the SOM under RBCI irrigation was also higher compared with FBCI. Guo et al. also found that the average soil organic carbon concentration and total nitrogen concentration under BW irrigation were higher than those under FW irrigation [4]. This may be due to successive BW irrigations weakening microbial activity, reducing microbial mineralization and inhibiting organic matter decomposition, thereby increasing the soil organic carbon content with continuous salinization [43].

### 3.2. Response of Secondary Soil Salinization to RBCI

Soil pH and ESP are two general indexes used throughout the world for dividing alkaline soil [44], and they are also the major elements that affect a soil’s dispersibility [45]. Overall, a pH that exceeds 8.5 and an ESP above 15% is alkaline soil. The results in this study showed that the pH and ESP values in R were higher than those in F, but they all did not exceed the threshold, and did not lead to secondary soil salinization. In previous research, the physicochemical properties and functions of soil were not influenced by short-term RW irrigation [46]. Our results indicated that the pH value under BW irrigation exceeded that under FW irrigation, which was due to the higher pH value in BW and the accumulation of the strong alkaline ion Na^+^; however, Guo et al. found that saline water irrigation reduced the soil pH value [4], which mainly depended on salinization and the accumulation of strong acidic ions, including NO_3_^−^, SO_4_^2−^ and Cl^−^. The salinity in irrigation water does not necessarily result in secondary soil salinization, but long-term utilization requires corresponding farming measures to prevent it [47]. This study found that the ESP under BW irrigation was over 15%, with a potential risk of alkalization, while the ESP did not exceed 15% under cyclic irrigation without a risk of soil alkalization. Guo et al. also found that there was no secondary salinization in soil that was treated with FBCI [24].

### 3.3. Response of Growth and Physiological Index of Crops to RBCI

Salt stress could alter the physiological and biochemical traits of different plants [48,49,50,51]. Plants can change their physiological and biochemical responses as a defense mechanism against salt stress [52,53]. It was found that salinity had adverse impacts on plant physiological and biochemical processes, thus diminishing their yield [54,55]. For example, salinity can restrain plant growth, photosynthesis, photosynthetic pigments, leaf water potential and result in cell swelling, damage cell membranes, accumulation of reactive oxygen species, and lead to nutritional and hormonal imbalances [56]. Neto et al. also found that salinity had an adverse impact on the primary physiological response of *Lippia alba* [57]. In general, salt stress leads to retrogressive development of the aboveground part of the plant, which is related to the defoliation partially mediated by ethylene production. Tanveer et al. (2020) found that salinity reduced the growth of tomato and significantly decreased the fresh weight and dry weight of the roots and aboveground biomass [58]. The results in this study also showed that BW irrigation declined crop biomass compared with freshwater irrigation, while reclaimed water irrigation increased crop biomass. This is because reclaimed water contains certain salts, but also contains high nutrient elements such as potassium, calcium and magnesium. In addition, RBCI can improve crop biomass compared to pure BW irrigation, because plants have the ability to form some type of “pressure memory”. This ability is defined as the ability of plants to store this information when they are initially stressed, so that they can respond differently to stress when they are again subjected to it. It usually triggers a more effective and faster response [59,60,61,62]. The chlorophyll content of leaves represents a photosynthetic rate that is sensitive to stress [63,64]. Salt stress may lead to the chlorophyll content increasing, which may lead to an increase in the number of chloroplasts per cell in the leaves of stressed plants [65,66]. that the chlorophyll content was found to decline significantly as the salinity increased [42,67]. The results in this experiment showed that compared with FW irrigation, BW irrigation increased the content of chlorophyll b, but decreased the chlorophyll a and total chlorophyll content.

### 3.4. Response of Antioxidant Enzymes of Leaves to RBCI

Salinity in the soil can diminish the water availability to plants and boost the change in water status, which directly affects the main physiological processes of plants. The biochemical responses to salt stress include the accumulation of organic and inorganic osmotic regulators, and improvements in the efficiencies of enzymatic and non-enzymatic antioxidant systems [40,68]. Therefore, compared with roots, the Na^+^ accumulation in leaves is less, delaying the toxic effects of salt, and this is an effective salt-tolerant mechanism [41]. Our results revealed that the Na^+^ content in leaves under cyclic irrigation was lower than that under pure BW irrigation, but that the Na^+^ content in leaves under RBCI was higher than that under FBCI, which may be due to some Na^+^ in reclaimed water.

The increase in MDA under salt stress depends on the ROS increasing and the damage to the cell membrane. Salt stress may cause a significant increase in MDA [42], and the MDA content was found to increase in tomato seedlings as the salinity increased [69]. The results of this experiment revealed that no significant difference was found in the MDA content between BW irrigation and FW irrigation, which may be caused by differences in crop species and the duration of salt stress. However, in our experiment, RBCI decreased the MDA content in leaves on the whole, indicating that it had a certain regulatory function in easing salt stress.

The accumulation of osmotic protective agents, such as sugars, proteins and some amino acids, leads to osmotic regulation, reduces the water potential of cells and maintains water absorption even in low-water-potential soils, such as saline soils [70]. In addition, these osmotic protective agents can also scavenge reactive oxygen species, stabilize proteins and cell membranes, and are important molecules of salt-tolerant stress [71,72]. Drought or salt stress was found to cause the proline content and protein content in lettuce to increase [73,74,75,76]. The strengthening of salt stress leads to increasing protein content [3], which is thought to help lettuce avoid oxidative damage. Analogous results were discovered in our experiment, namely that the content of soluble protein in leaves under BW irrigation was higher compared to that under FW irrigation. Compared with pure BW irrigation, the leaf protein content under RBCI increased at a lower salinity in BW (3 g·L^−1^), but decreased at a higher salinity in BW (5 g·L^−1^).

Superoxide dismutase (SOD) protects plants from oxidative damage by catalyzing the conversion of O_2_^−^ to H_2_O_2_ [77]. Reactive oxygen species (ROS) were detected by POD [78], while CAT catalyzed the decomposition of hydrogen peroxide into H_2_O and O_2_ to reduce ROS levels [79]. The results of this study indicated that there was no significant change in the activities of POD and CAT in leaves under BW irrigation compared with FW irrigation. Similarly to our results, the water-salt stress did not increase the activity of defensive enzymes involved in oxidative stress, such as SOD and POD [3], and did not induce SOD under a salt stress of 40 mM [80]. However, some results were inconsistent, such as the strengthening of water and salt stress significantly increasing CAT activity [3]. This may be caused by differences in water stress, which was not involved in this experiment.

## 4. Materials and Methods

### 4.1. Tested Soil

The tested soil was taken from the topsoil in a field near the Xinxiang Agricultural soil and Water Environment Field Scientific observation Station of the Chinese Academy of Agricultural Sciences. The soil was dried, crushed and screened (2 mm). The soil bulk density was 1.40 g·cm^−3^, the field capacity was 17.27%, the total nitrogen content was 0.668 g·kg^−1^, the total phosphorus content was 0.385 g·kg^−1^, the soil–water specific conductivity was 0.264 dS·m^−1^, and the organic matter content was 2.31%. The BT-9300HT laser particle size analyzer was used to analyze the soil sample particles. The clay (<0.002 mm), silt (0.002~0.02 mm) and sand (0.02~2 mm) accounted for 20.90 percent, 44.62 percent and 34.48 percent, respectively, and the soil texture belonged to loams (international system).

### 4.2. Experimental Device and Scheme

The experiment began in October and ended in December 2020, in the greenhouse of the station. Shanghai green was selected as the tested crop. The pot experiment involved setting up two factors, including the salinity of BW and irrigation sequence, in which the salinity in the BW had two levels (3 and 5 g·L^−1^), the irrigation cycle sequence was established with four levels (BW, RW-BW, RW-RW-BW, and RW), and FBCI was used as the control group. The experimental design was carried out in 12 treatments with 3 repetitions (Table 5). The upper diameter, lower diameter and height of the plastic pot were 25, 14.5 and 19 cm, respectively, and 3 small holes were punched at the bottom. A mass of 7 kg of soil was placed in each pot, and the compound fertilizer (N-P_2_O_5_-K_2_O of 15-15-15) dosage used was 1 g per 1 kg soil, referring to the local fertilizer application rate. Sowing occurred on 9 October 2020, and 5 seedlings were kept in each pot at the two-leaf stage (31 October) to begin the watering treatment. Traditional surface irrigation was adopted when the soil water content was below 75 percent of the field’s capacity, and irrigation amount was approximately 300 mL. The soil moisture was monitored using a portable soil moisture meter. No drainage occurred during the growing period. The qualities of the FW, BW and RW are shown in Table 6. The RW was taken from the Luotuowan Domestic Sewage Treatment Plant in Xinxiang City, Henan Province. The plant adopted a A^2^/O treatment process, and the treated water quality met the “Farmland Irrigation Water quality Standard” (GB5084-2021). The FW was tap water, and the BW was obtained by adding sea salt into the FW according to the results of [81]. In the experiment, except for the different irrigation sources, the other conditions were consistent, such as the irrigation amount, irrigation method, irrigation time, fertilization amount, agronomic measures, etc.

### 4.3. Measured Indices and Methods

(1)Physicochemical properties of soil. After crop harvest, soil samples were collected and then air-dried, ground and sieved (2 mm). The soil moisture content was measured with the oven-drying method. The soil sample was extracted with a soil-to-water ratio of 1:5. Then, the extracts were used to measure the electrical conductivity (EC) with a conductivity meter, Na^+^ by flame photometry, Cl^−^ by AgNO_3_ titration, and the soil organic matter (SOM) was determined using a low-temperature external-heat potassium dichromate oxidation-colorimetric method according to the methods of [82]; the water drop penetration time (WDPT) was measured via the water drop penetration time method according to the methods of [22].(2)Soil salinization index. The soil pH, exchangeable ions, exchangeable soil sodium percentage (ESP) and effective action exchange capacity (ECEC) were determined and calculated according to the methods of [83].(3)Soil enzyme activity. The activities of soil alkaline phosphatase (S-AKP/ALP) were determined with detection kits (Solarbio, Beijing), and the daily release of 1 nmol phenol per gram of soil at 37 °C was used as an enzyme activity unit. The soil sucrase (S-SC) activity was determined using 3,5-dinitrosalicylic acid colorimetry, and its activity was expressed as milligrams of 1 g of soil glucose after 24 h. The activity of soil urease (S-UE) was determined with indophenol-blue colorimetry, and its activity was expressed by the number of milligrams of NH_3_-N in 1 g of soil after 24 h.(4)Growth and physiological index of crop. After harvesting (14 December), the aboveground fresh weight (AFW), aboveground dry weight (ADW), underground fresh weight (UFW) and underground dry weight (UDW) were determined, referring to the methods described in [84]. A detection kit (Suolebao, Beijing) was used to measure the leaf chlorophyll contents.(5)Antioxidant index of crop. Using methods outlined in [84], the soluble protein content, catalase (CAT), superoxide dismutase (SOD), peroxidase (POD) activities and malondialdehyde (MDA) content were determined.(6)Na^+^ content in leaves. The Na^+^ content in leaves was determined via a flame photometer method.

### 4.4. Data Analysis

The experimental data were calculated using Excel 2010. A multivariate analysis of variance was used to analyze the differences among treatments, using SPSS25.0 software (IBM Crop., Armonk, NY, USA). The significance level was set to 0.05. The figures were illustrated using origin 2019b software.

## 5. Conclusions

In areas where BW resources are distributed, the available amounts of freshwater resources are generally relatively low, resulting in less water used for agricultural irrigation. This limits the application of brackish water-freshwater cyclic irrigation to some extent, thereby affecting the utilization of brackish water. China has a large amount of reclaimed water resources, but the development and utilization rate of these resources is low. Reclaimed water has a lower salt content compared to brackish water, and can replace freshwater to leach salt. We studied the effects of brackish water and reclaimed water wheel irrigation on soil and crops through pot experiments, and obtained the main conclusions, as follows:
(1)Compared with FBCI, the soil water content increased without a significant level, while the soil EC, sodium and chloride ions contents improved obviously under RBCI. The contents of soil EC, sodium and chloride ions gradually declined, while the soil moisture content decreased gradually as irrigation times were increased using reclaimed water under RBCI.(2)The responses of different soil enzyme activities to RBCI were different. As irrigation times increased using reclaimed water, the difference in S-UE activity reached a significant level; for the same irrigation sequence, the activities of soil alkaline phosphatase, sucrase and urease increased with the increase in salinity in brackish water.(3)RBCI can alleviate the risk of secondary soil salinization to some extent. The soil pH values were all below 8.5 without any soil alkalization risk. The ESP did not exceed 15%, and there was no possible risk of soil alkalization except that the ESP of B3 and B5 went beyond the limit of 15%.(4)Neither the aboveground or underground biomass reached an obvious difference between FBCI and RBCI. RBCI was conducive to improving the aboveground biomass compared to irrigating crops with brackish water.(5)Cyclic irrigation using reclaimed-reclaimed-brackish water at 3 g·L^−1^ was recommended under the experimental conditions.

The irrigation amount, especially of reclaimed water, is important to salt leakage. In this experiment, only two levels in salinity of brackish water and one kind of reclaimed water were considered, while the irrigation amount was not considered. In order to consider the irrigation amount and the type of reclaimed water, further experimental research is needed. In addition, for the pot experiment, the distribution of salt in the soil profile could not be reflected due to the limited height of the pot. In the future, long-term field experiments should be carried out.

## Figures and Tables

**Figure 1 plants-12-02285-f001:**
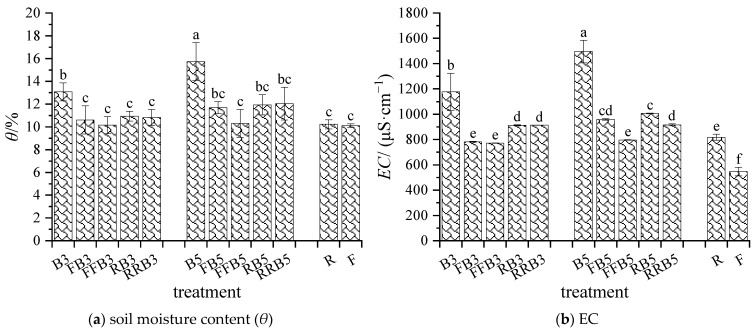
Variations in soil water and salt content under cyclic irrigation. (**a**) soil moisture content (*θ*) under different treatments after harvest; (**b**) EC in soil extract under different treatments after harvest. Note: different lowercase letters on the bars represent significant differences at the level of 0.05.

**Figure 2 plants-12-02285-f002:**
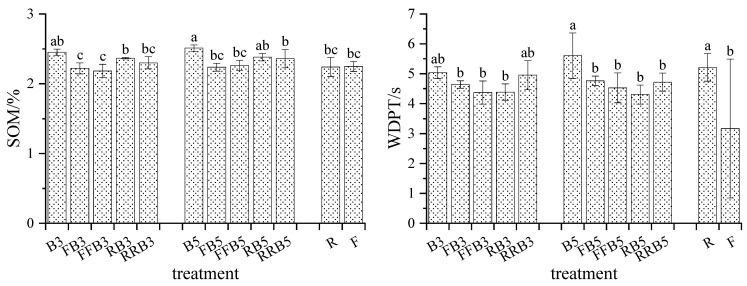
Variations in SOM and WDPT under cyclic irrigation. Note: different lowercase letters on the bars represent significant differences at the level of 0.05.

**Figure 3 plants-12-02285-f003:**
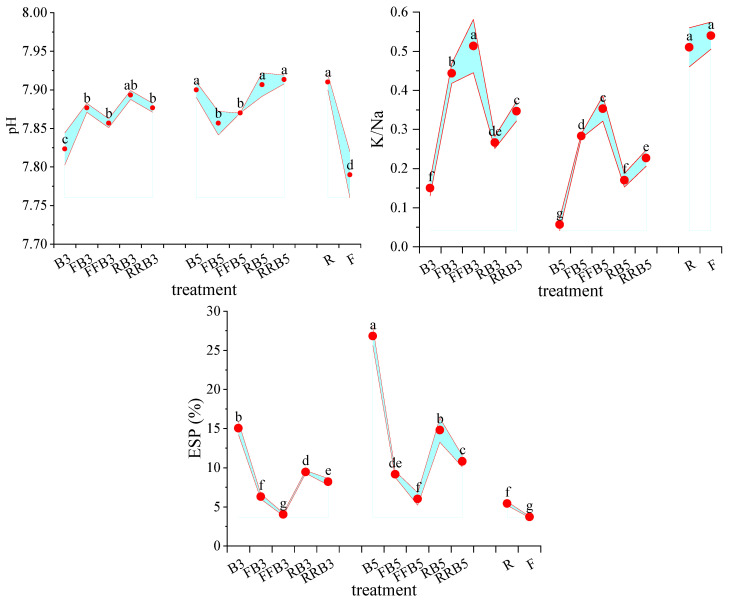
Variations in soil pH value, K/Na, ESP and SAR under cyclic irrigation. Note: different lowercase letters on the points represent significant differences at the level of 0.05.

**Figure 4 plants-12-02285-f004:**
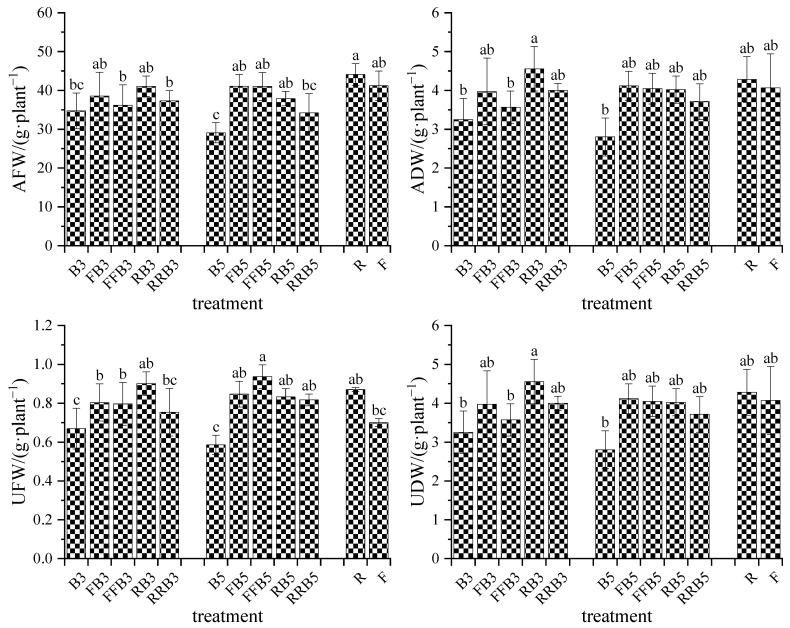
Variations in biomass of pakchoi under cyclic irrigation. Note: different lowercase letters on the bars represent significant differences at the level of 0.05.

**Figure 5 plants-12-02285-f005:**
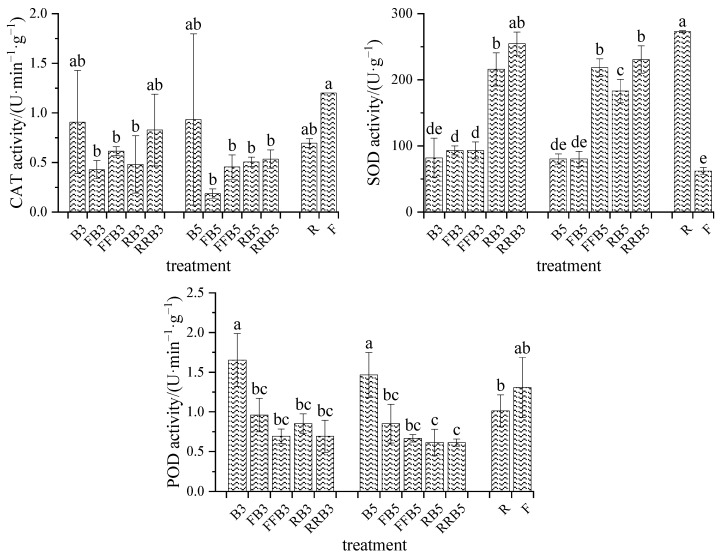
Variations in enzyme activities of leaves under cyclic irrigation. Note: different lowercase letters on the bars represent significant differences at the level of 0.05.

**Figure 6 plants-12-02285-f006:**
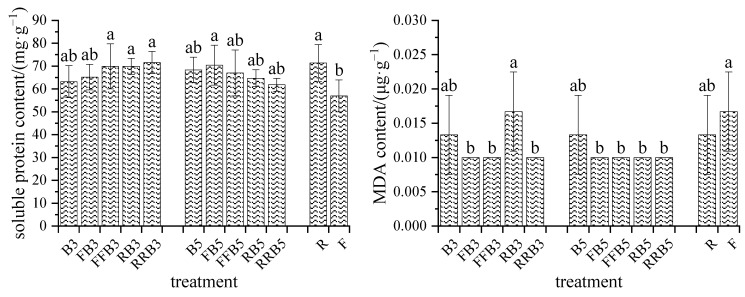
Variations in MDA and soluble protein contents of leaves under cyclic irrigation. Note: different lowercase letters on the bars represent significant differences at the level of 0.05.

**Figure 7 plants-12-02285-f007:**
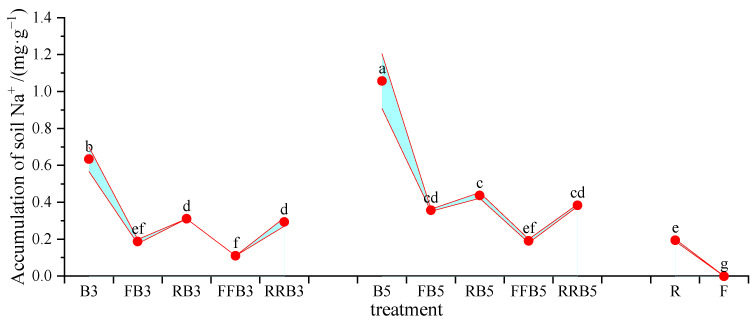
Accumulations of Na^+^ content in soil under different irrigation rotation methods. Note: different lowercase letters on the points represent significant differences at the level of 0.05.

**Figure 8 plants-12-02285-f008:**
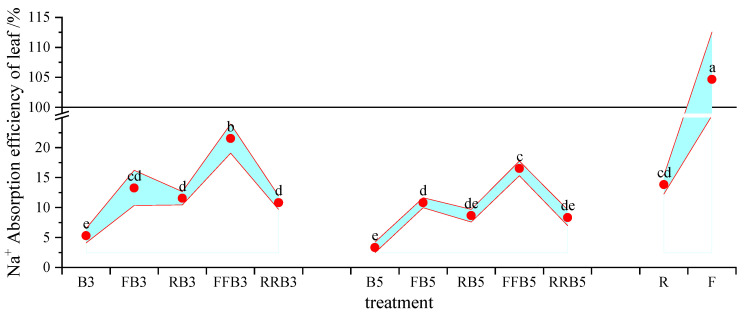
Na^+^ absorption efficiency of leaves under different cyclic irrigation methods. Note: different lowercase letters on the points represent significant differences at the level of 0.05.

**Table 1 plants-12-02285-t001:** Variations in soil ion contents after cyclic irrigation.

Treatment	Na^+^ Content/(mg·kg^−1^)	Cl^−^ Content/(mg·kg^−1^)
B3	676.67 ± 68.25 b	175.92 ± 17.68 b
FB3	230.00 ± 8.66 ef	96.90 ± 0.68 ef
RB3	351.67 ± 2.89 d	112.55 ± 1.33 de
FFB3	152.17 ± 0.76 f	95.71 ± 0.77 ef
RRB3	338.33 ± 23.09 d	107.83 ± 0.93 de
B5	1098.33 ± 146.32 a	257.75 ± 9.57 a
FB5	398.33 ± 7.64 d	131.31 ± 1.79 c
RB5	480.00 ± 13.23 c	132.79 ± 1.11 c
FFB5	231.67 ± 7.64 ef	105.61 ± 2.71 e
RRB5	425.00 ± 5.00 cd	118.61 ± 3.58 d
R	235.00 ± 5.00 e	93.21 ± 7.29 f
F	43.67 ± 1.04 g	61.74 ± 3.69 g

Note: different lowercase letters behind data in the same column represent significant differences at the level of 0.05.

**Table 2 plants-12-02285-t002:** Variations in soil enzyme activities under cyclic irrigation.

Treatment	S-AKP/ALP Activity /(U·g^−1^)	S-SC Activity /(mg·g^−1^·24 h^−1^)	S-UE Activity /(mg·g^−1^·24 h^−1^)
B3	2105.95 ± 344.81 bc	9.59 ± 0.40 c	0.39 ± 0.01 e
FB3	1881.55 ± 431.21 c	11.37 ± 0.44 ab	0.46 ± 0.00 c
RB3	2744.64 ± 719.43 b	10.73 ± 0.26 bc	0.42 ± 0.01 d
FFB3	2641.07 ± 773.33 bc	10.91 ± 0.64 bc	0.45 ± 0.01 c
RRB3	2623.81 ± 837.16 bc	12.16 ± 1.16 a	0.44 ± 0.01 c
B5	2313.09 ± 367.4 bc	10.43 ± 0.55 bc	0.39 ± 0.01 e
FB5	3935.71 ± 499.41 a	10.04 ± 0.84 bc	0.48 ± 0.01 b
RB5	3521.43 ± 404.46 ab	11.69 ± 0.41 ab	0.45 ± 0.01 c
FFB5	2364.88 ± 239.18 bc	11.71 ± 0.68 ab	0.48 ± 0.01 b
RRB5	2606.55 ± 374.63 bc	10.92 ± 0.89 bc	0.51 ± 0.02 a
R	3901.19 ± 528.96 a	11.01 ± 0.39 b	0.45 ± 0.02 c
F	3348.81 ± 29.9 ab	10.86 ± 1.13 bc	0.41 ± 0.01 de

Note: different lowercase letters behind data in the same column represent significant differences at the level of 0.05.

**Table 3 plants-12-02285-t003:** Variations in chlorophyll contents of pakchoi under cyclic irrigation.

Treatment	Content of Chlorophyll a/(mg·g^−1^)	Content of Chlorophyll b/(mg·g^−1^)	Content of Total Chlorophyll/(mg·g^−1^)
B3	1.67 ± 0.18 ab	0.43 ± 0.01 cd	2.1 ± 0.18 ab
FB3	1.5 ± 0.08 b	0.47 ± 0.02 c	1.96 ± 0.1 b
RB3	1.92 ± 0.14 a	0.52 ± 0.02 b	2.44 ± 0.13 a
FFB3	1.59 ± 0.08 ab	0.5 ± 0.02 bc	2.08 ± 0.1 ab
RRB3	1.73 ± 0.03 ab	0.56 ± 0.07 a	2.29 ± 0.1 ab
B5	1.78 ± 0.11 ab	0.44 ± 0.01 cd	2.22 ± 0.1 ab
FB5	1.65 ± 0.07 ab	0.49 ± 0.02 bc	2.14 ± 0.09 ab
RB5	1.78 ± 0.29 ab	0.47 ± 0.02 c	2.25 ± 0.29 ab
FFB5	1.46 ± 0.07 b	0.46 ± 0.02 c	1.92 ± 0.09 b
RRB5	1.77 ± 0.3 ab	0.45 ± 0.02 cd	2.21 ± 0.29 ab
R	1.62 ± 0.38 ab	0.44 ± 0.02 cd	2.07 ± 0.39 b
F	1.85 ± 0.26 a	0.41 ± 0.03 d	2.26 ± 0.25 ab

Note: different lowercase letters behind data in the same column represent the significant differences at the level of 0.05.

**Table 4 plants-12-02285-t004:** Variations in Na^+^ content of soil and leaves under cyclic irrigation.

Treatment	Na^+^ Content in Soil/(mg·g^−1^)	Na^+^ Content in Leaf/(mg·g^−1^)
B3	0.68 ± 0.0683 b	15.07 ± 0.26 b
FB3	0.23 ± 0.0087 ef	10.01 ± 0.04 e
RB3	0.35 ± 0.0029 d	12.36 ± 0.33 cd
FFB3	0.15 ± 0.0008 f	11.60 ± 0.43 d
RRB3	0.34 ± 0.0231 d	12.43 ± 0.25 cd
B5	1.1 ± 0.1463 a	17.85 ± 1.26 a
FB5	0.4 ± 0.0076 cd	14.63 ± 0.54 b
RB5	0.48 ± 0.0132 c	14.34 ± 1.06 b
FFB5	0.23 ± 0.0076 ef	12.86 ± 0.83 c
RRB5	0.43 ± 0.005 cd	13.01 ± 0.67 c
R	0.24 ± 0.005 e	10.02 ± 0.18 e
F	0.04 ± 0.001 g	6.85 ± 0.73 f

Note: different lowercase letters behind the data in the same column represent significant differences at the level of 0.05.

**Table 5 plants-12-02285-t005:** Design for cyclic irrigation using different water sources.

Treatment	B3	FB3	FFB3	B5	FB5	FFB5	F	RB3	RRB3	RB5	RRB5	R
Irrigation water	BW3	FW- BW3	FW-FW- BW3	BW5	FW- BW5	FW-FW- BW5	FW	RW- BW3	RW-RW- BW3	RW- BW5	RW-RW- BW5	RW

Note: The 3 and 5 represent 3 and 5 g·L^−1^ of salinity in BW, respectively.

**Table 6 plants-12-02285-t006:** Water quality of RW, BW and FW.

Water Source	EC	pH	Na^+^	K^+^	HCO_3_^−^	Cl^−^	Ca^2+^	Mg^2+^	SO_4_^2−^	SAR	TN	TP	Pb	Cu	Zn	Cd
FW	321	8.31	0.4	0.04	1.96	0.85	0.98	0.61	1.08	0.34	1.17	0.02	-	-	-	-
RW	2120	8.17	13.5	0.36	4.56	8.85	2.28	3.10	5.28	5.81	0.52	0.05	-	-	-	-
BW3	6100	8.41	57.8	0.05	2.32	54.20	1.08	0.71	0.96	43.21	1.31	0.02	-	-	-	-
BW5	9432	8.44	87.0	0.07	2.28	90.90	0.92	0.77	1.14	66.86	1.18	0.02	-	-	-	-

Note: EC represents electrical conductivity, μS·m^−1^; SAR represents sodium adsorption ratio, (mmol·L^−1^)^0.5^; TN represents total nitrogen content, mg·L^−1^; TP represents total phosphorus content, mg·L^−1^; for Pb, Cu, Zn and Cd, the units are mg·L^−1^; for Na^+^, K^+^, HCO_3_^−^, Cl^−^, Ca^2+^, Mg^2+^ and SO_4_^2−^, the units are mmol·L^−1^; “-” indicates not detected: concentration was below the instrumental detection limit.

## Data Availability

The data used to support the findings of this study are available from the corresponding author upon request.

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
