# Peer review of "Response of Soil Microenvironment and Crop Growth to Cyclic Irrigation Using Reclaimed Water and Brackish Water"

_plants, 2023, doi:10.3390/plants12122285_

Round 1

Reviewer 1 Report

Your manuscript requires language revisions.

Author Response

In the abstract,
Please be more precise, among the different treatments applied, which one gives the best result?

Reply. Yes, you are right. We add the more precise result at the end of the abstract: Cycle irrigation using reclaimed-reclaimed- brackish water of 3 g·L-1 was recommended according to the experimental results.

Keywords - please put in alphabetical order.

Reply: Thanks for your advice. The keywords have been put in alphabetical order.

Line 50- 54
This sentence needs a reference

Reply: Thanks for your advice. We have added a reference for the sentence.

Line 55
Add a space between the sentence in the reference (The same remark in all the document)
At the end of Introduction
-Provide clear testable research hypotheses, tell the assumed mechanism of the processes, tell
what were your expectations.
Show the link between hypothesis – dataset used for testing – statistical test selected for
proof.

Reply: Thanks for your advice. We have added space between the sentence and the reference. At the end of introduction, we add the testable research hypotheses. We hypothesized that reclaimed water could replace freshwater to irrigate crop with brackish water alternatively. Our objective was to identify the cycle irrigation mode using brackish water and reclaimed based on crop characteristics and soil environmental indictors, and to provide some guidance for marginal quality water using for agricultural irrigation in freshwater-scarce areas.

Line 408
Add reference date

Reply: Thanks for your advice. We have added the reference date.

Lines 438-448
This part should be more developed and enriched.
In the discussion I propose to start with the MDA stress indicator, and then osmolytes and
antioxidant enzymes.

Reply: Thanks for your advice. We accept your advice, and adjust the order: start with the MDA stress indicator, and then osmolytes and antioxidant enzymes.

Line 494
Add space between the number and unit. The same remark in all the document.

Reply: Thanks for your advice. We have added space between the number and unit.

Line 504
What do you mean by potential by A2/O

Reply: The A2/O process is a combination of anaerobic/aerobic phosphorus removal systems and anaerobic/aerobic nitrogen removal systems. It is the basic process for biological nitrogen and phosphorus removal, and can simultaneously remove BOD, nitrogen, and phosphorus from water.

Line 506
It is necessary to indicate the reference of all the analyses used

Reply: Yes, you are right. We added the reference.

line 511
Table titles must be consistent in writing style

Reply: Yes, you are right. We keep the table titles consistent in writing style.

Line 561
Specify the type of ANOVA that has been performed

Reply: Thanks for your advice. The type of ANOVA is multivariate analysis of Variance, which has been added in the revision.

Conclusion
Should be redrafted and clearly emphasized

Reply: Thanks for your advice. We revised the conclusion.

References
As concern references, the way of citing references should be uniform
Please verify the entire manuscript again in terms of compliance with the template required
by the Plants journal.
Your manuscript requires language revisions.

Reply: Thanks for your advice. We revised according to the template. And The language has been polished by doctor Cui. He is a reviewer for multiple journals, such as Agronomy, Water, Science of the total environment, etc.

Reviewer 2 Report

Dear authors,

very nice overall work.

I would like to see some more information about the methodology that was not very clearly stated.

plus, there is no mention about limitations and future research..

Last, there are some relevant refernces for irrigation demand in Mediterranean area (Spain, Italy, Greece, Cyprus), that can also help the context of your research.

Best

Please check the language for some syntax mistakes.

Author Response

I would like to see some more information about the methodology that was not very clearly stated.

plus, there is no mention about limitations and future research..

Last, there are some relevant references for irrigation demand in Mediterranean area (Spain, Italy, Greece, Cyprus), that can also help the context of your research.

Reply: Thanks for your advice. We add some contents about the methodology, limitations and future research for our topic. Irrigation amount, especially reclaimed water, is import to the salt leaking. In this experiment, only two levels in salinity of brackish water and one kind of reclaimed water were considered, and irrigation amount were not considered. In order to consider the irrigation amount and the type of reclaimed water, further experimental research is needed. In addition, the pot experiment was used in the experiment, but the distribution of salt in the soil profile could not be reflected due to the limited height of the pot. In the future, long-term field experiment should be carried out.

We will carefully read some relevant references for irrigation demand in Mediterranean area (Spain, Italy, Greece, Cyprus) to improve our research.

Reviewer 3 Report

COMMENTS TO THE AUTHORS

Dear Authors

The text is clear and interesting. However some modifications should be done.

1) I recommend that the section “Material and methods” should be placed before the section “Results”.

2) Material and methods

a) The number of replications of each treatment is low (only three); please, at least, take care to the next research plan (now is too late).

b) Irrigation systems, water distribution uniformity and irrigation efficiency should be presented. The main reason is because the irrigation efficiency is the ratio between the water stored in the plant root zone and the water applied during the irrigation (%).  The loss of irrigation water is due to the non-uniformity distribution of water, evaporation during irrigation (and not percolation on the present experiments once that there is no drainage water below the root zone).

c) Irrigation with treated effluent is a promising technology to simultaneously satisfy water and nutrients demand and avoid pollution problems. However, if the treated wastewater is reused microbial and heavy metals should be mentioned.

d) Additional ions (treated effluents) – nitrates, potassium ammonium and phosphorus should also determined. This is important, according to their influence on crop growth and yield, and on the increase of salt tolerance due to the presence of potassium and nitrogen ions.

e ) You should mention soil type of the experimental site.

3) According to the referred new data of the section “Material and methods”, if needed, you should refer its influence on the sections Results, Discussion and Conclusions.

4) Conclusions - I think that adding few lines dealing with importance of the manuscript topic is desirable.

Author Response

1) I recommend that the section “Material and methods” should be placed before the section “Results”.

Reply: Thanks for your advice. According to the template of “Plants”, “Material and methods” is placed after the section “Results”.

2) Material and methods

  1. a) The number of replications of each treatment is low (only three); please, at least, take care to the next research plan (now is too late).

Reply: Yes, you are right. We will increase the number of replications of each treatment in future research.

  1. b) Irrigation systems, water distribution uniformity and irrigation efficiency should be presented. The main reason is because the irrigation efficiency is the ratio between the water stored in the plant root zone and the water applied during the irrigation (%). The loss of irrigation water is due to the non-uniformity distribution of water, evaporation during irrigation (and not percolation on the present experiments once that there is no drainage water below the root zone).

Reply: Thanks for your advice. In the experiment, The traditional surface irrigation was used to irrigate crop when the soil water content was lower than 75% of field capacity, and irrigation amount was about 300 mL. In the experiment, except for the different irrigation sources, other conditions were consistent, such as irrigation amount, irrigation method, irrigation time, fertilization amount, agronomic measures, etc.

  1. c) Irrigation with treated effluent is a promising technology to simultaneously satisfy water and nutrients demand and avoid pollution problems. However, if the treated wastewater is reused microbial and heavy metals should be mentioned.

Reply: Thanks for your advice. As shown in Table 6, there was no heavy metals in irrigated water, including freshwater, brackish water, and reclaimed water (treated effluent), so we did not measured the heavy metals in soil or plant.

  1. d) Additional ions (treated effluents) – nitrates, potassium ammonium and phosphorus should also determined. This is important, according to their influence on crop growth and yield, and on the increase of salt tolerance due to the presence of potassium and nitrogen ions.

Reply: Thanks for your advice. In treated effluents (reclaimed water), we determined the potassium, total nitrogen, total phosphorus, as shown in table 6.

e ) You should mention soil type of the experimental site.

Reply: Thanks for your advice. We mentioned the soil type in 4.1 section “the soil texture belongs to loam (international system)”. Nitrate nitrogen and ammonium nitrogen will be measured in the future research.

3) According to the referred new data of the section “Material and methods”, if needed, you should refer its influence on the sections Results, Discussion and Conclusions.

Yes, you are right. We added some contents in discussion.

4) Conclusions - I think that adding few lines dealing with importance of the manuscript topic is desirable.

Yes, you are right. We add some lines dealing with the important of our topic.